# Physical Activity Levels in Brazilian Adolescents: A Secular Trend Study (2007–2017/18)

**DOI:** 10.3390/ijerph192416901

**Published:** 2022-12-16

**Authors:** André Araújo Pinto, Rômulo Araújo Fernandes, Kelly Samara da Silva, Diego Augusto Santos Silva, Thais Silva Beltrame, Fernando Luiz Cardoso, Andreia Pelegrini

**Affiliations:** 1Department of Physical Education, State University of Roraima, Sete de Setembro Street, 231, Canarinho, Boa Vista 69306-530, Brazil; 2Laboratory of InVestigation in Exercise (LIVE), Department of Physical Education, Sao Paulo State University (UNESP), Presidente Prudente 19060-900, Brazil; 3Centre in Physical Activity and Health, Federal University of Santa Catarina, Florianópolis 88040-400, Brazil; 4Research Center in Kinanthropometry and Human Performance, Federal University of Santa Catarina, Florianópolis 88040-400, Brazil; 5Faculty of Health Sciences, Universidad Autónoma de Chile, Providencia 7500912, Chile; 6Center of Health and Sports Sciences, University of Santa Catarina State, Pascoal Simone Street, 358, Coqueiros, Florianópolis 88080-350, Brazil; 7Study and Research Group in Kinanthropometry, Center of Health and Sports Sciences, University of Santa Catarina State, Pascoal Simone Street, 358, Coqueiros, Florianópolis 88080-350, Brazil

**Keywords:** adolescence, exercise, health, trends

## Abstract

Only a small proportion of Brazilian adolescents practice sufficient physical activity (PA). However, it is not clear whether this proportion has been decreasing over time. This study aimed to examine the 10-year trends of sufficient PA in adolescents and to investigate differences by sex and age. Using a standard protocol, we compared two cross-sectional cohorts of adolescents aged 15 to 18 years, recruited in 2007 (n = 1040) and in 2017/18 (n = 978). Using the International Physical Activity Questionnaire—Short Form (IPAQ-SF), the adolescents reported moderate-to-vigorous PA (MVPA) performed in the last seven days. Sufficient PA was defined as engaging in at least 60 min/day of MVPA. In the subgroups, investigated by sex or age, there was an increase in the prevalence of sufficient PA. Overall, sufficient PA declined by 28.1% from 2007 to 2017/18. Boys in 2007 were more active than their 2017/18 peers, and this was equally observed in girls in 2007 compared to those in 2017/18. The findings show decreasing secular trends in sufficient PA in the investigated adolescents. Not only are public health authorities in Brazil witnessing an escalation of insufficient PA, but they are also losing ground with the most active adolescents.

## 1. Introduction

Sufficient levels of physical activity (PA) provide improvements in physical fitness and bone health, reduce the risk of developing chronic diseases, and are crucial for weight control [1]. During adolescence, if sufficient PA levels are sustained, the benefits may last into adulthood [2]. This does not seem to be an easy thing to achieve, since global data revealed that 81.0% of adolescents do not meet the 60 min daily of moderate-to-vigorous PA (MVPA) suggested by the World Health Organization [1,3]. This scenario has motivated the conduction of several studies around the world in search of evidence to prove the decline in sufficient PA over time in adolescents [4,5,6]. Elegant studies investigating the secular trends of sufficient PA in adolescents have indicated results that go in different directions, sometimes indicating a decline in PA [4,7], an increase [5,8], and stability [6,9]. When compiled, the studies provided data mainly based on self-reports, from different PA contexts, showing trends to increase participation in sports activities, and with little information from low- and middle-income countries [10].

In addition, it is unclear whether trends differ between sexes and age regardless of their direction (decline, increase, or stability). For example, trends of increasing sufficient PA in boys and stable PA in girls were observed in a global report with data collected in 2001 and 2016 [3]. In Norway, sufficient PA decreased between 2005 and 2012 in girls and remained stable in boys over the same period [4]. Another study comparing PA in adolescents from the Czech Republic between 2006 and 2014 showed declining trends in both sexes [7]. In Canada, a study comparing data from 2005 and 2014 showed a trend of stability in PA for both sexes [11]. Regarding age, while PA declined in Portugal between 2006 and 2016 in younger adolescents [12], in Norway and Finland, PA increased in younger adolescents between 1985 and 2014 [8].

In Brazil, which is characterized as a middle-income country, data based on the International Physical Activity Questionnaire (IPAQ) showed that the prevalence of sufficient PA ranges from 3.3% to 77.7% [13]. At the national level, data on PA in adolescents started to be collected in 2009 by the Pesquisa Nacional de Saúde do Escolar (PeNSE) (free translation: National School Health Survey) [6]. However, some modifications in the research protocols have prevented the analysis of secular trends in PA in Brazilian adolescents. Due to these changes, a study with data from PeNSE, using only comparable information from leisure PA, revealed a small upward trend between 2009 (13.8%) and 2015 (14.8%) [14]. The authors advised of the importance of exploring PA in the regions, as Brazil is a country with continental dimensions, with different levels of economic equality, climate, and lifestyle. Additionally, trends in specific contexts, such as sports participation and commuting PA, were reported in a previous review study but not in habitual PA [15]. Thus, the unavailability of data reporting trends in habitual PA in Brazilian adolescents, even from specific regions, justifies the need for future investigations.

Given this context, we hypothesize that sufficient PA has declined over the years in Brazilian adolescents. To prove our hypothesis, data on secular trends in sufficient PA in Brazilian adolescents may provide a better understanding of the possible changes in PA. Moreover, these studies are crucial for public health because they provide information on the impact of public policies aimed at promoting PA in this population subgroup. With this information, it will be possible to direct strategies for PA promotion to improve this behavior among adolescents and to identify which subgroups are most in need of intervention. Thus, this study aimed to examine the 10-year trends of sufficient PA in adolescents and to investigate differences by sex and age.

## 2. Materials and Methods

### 2.1. Study Design and Sample

This study is based on the survey “Levels of physical activity, physical fitness, and social behavior related to adolescent health: a secular trend study” conducted in Florianopolis, Santa Catarina, Brazil. The survey includes cross-sectional information systematically collected every 10 years since 2007. Between August and December, data were collected from adolescents aged 15 to 18 years old. All study participants were recruited in a public education system managed by the Santa Catarina State. Data were collected at two different times (2007 and 2017/18). For logistical reasons, it was not possible to carry out the data collection in its entirety in the second half of 2017, so it was extended to the months of March to May in 2018.

According to the information from the school census provided by the Department of Education State, the enrolled adolescent population was 12,741 inhabitants in 2007 and 10,192 in 2017. In both surveys, the sample was determined following the same sampling parameter [16] using the same criteria described elsewhere in detail [17]. An additional 10% was established to mitigate any losses, reaching a minimum sample of 631 adolescents in 2007 and 624 in 2017.

Following the criteria of the first survey, the estimated sample number was distributed proportionally in the five regions organized according to the Municipal Health Secretary (Center, Continent, East, North, and South). The school with the highest number of enrollments in each region was selected for data collection. Then, the students were selected, drawing the required number of classes until the number of subjects required for that region was researched (saturation sampling). All students in the visited rooms were invited to participate (cluster sampling).

### 2.2. Inclusion Criteria

The adolescents provided written consent, which was also signed by their parents or legal guardians, authorizing their participation in the study. Adolescents with physical disabilities or psychological disorders were excluded during data screening. Adolescents aged 14 and under and those aged 19 and over were excluded due to the low number. The study protocol was approved by the National Health Council of Brazil in 2007 (protocol 372/2006) and 2017 (protocol 2,172,699).

### 2.3. Procedures and Measures

The same researcher, maintaining the standardization of data collection, including a self-administered questionnaire, managed the fieldwork in the two collection periods. Trained field professionals obtained all information. From the questionnaires, we obtained sociodemographic information, such as sex (boy; girl) and age in full years.

Socioeconomic status was assessed using the most updated version of a national indicator that includes several items of property in the adolescents’ homes both in 2007 and in 2017/18 [18]. In 2007, the items included, for example, color television, radio, and video cassette, and for each answer obtained, a score was generated, with the total score varying from 0 to 34. In 2017/18, other items were integrated, such as a DVD player, microcomputer, and microwave, with scores ranging from 0 to 100. Due to the alteration in the questionnaire, the socioeconomic status of the adolescents was classified into socioeconomic strata, namely, low (1st tertile), medium (2nd tertile), and high (3rd tertile), and it was used for characterization purposes.

PA was measured using the short version (for adolescent Brazilians) of the International Physical Activity Questionnaire (IPAQ) because it is recommended for the national monitoring of PA level [19]. The IPAQ-Short assesses the participation in all kinds of moderate-to-vigorous activities across various physical activity domains (i.e., leisure time, occupational, domestic, and transport), using the “last 7 days” as a reference period. The adolescents were required to recall the type, frequency (days per week), and duration (hours and minutes per day) of each activity performed during the last seven days. In this study, sufficient PA was classified based on meeting the current PA guidelines of at least 60 min of MVPA daily of the World Health Organization [1].

### 2.4. Statistical Analyses

Descriptive data are presented in percentages (%), divided by sex and age for each year. The prevalence of sufficient PA (≥60 min of MVPA per day) was calculated by dividing the number of adolescents who met the recommendations by the total number of adolescents multiplied by 100. The significant differences in sufficient PA between adolescents in 2007 and in 2017/18 can be seen by the non-overlapping of the 95% confidence intervals (95% CIs). In addition, the percentage changes were calculated as follows: (final value–initial value)/initial value × 100). Secular trends in sex and age were determined using binary logistic regression (Enter method) using 2007 data as a reference. We estimated the odds ratio and the 95% CI of sufficient PA (for each sex and age) in 2017/18. All data were inputted into the statistical software package IBM SPSS Statistics (Version 20.0; IBM Corp., Armonk, NY, USA). A level of significance of *p* < 0.05 was set for all the analyses.

## 3. Results

Table 1 shows a description of the sample considering the year of survey, sex, and age. In 2007, the total number of adolescents recruited was 1040 (404 boys and 636 girls). In 2017/18, 978 adolescents were interviewed (503 boys and 475 girls). Thus, the total sample included consists of the analysis of self-reported data from 2018 adolescents. The proportion of adolescents in each economic status was similar between periods.

Table 2 shows the prevalence of and variation in sufficient PA among adolescents in 2007 and 2017/2018. Overall, adolescents in 2007 (50.8%; 95% CI = 47.1–52.4) were more active than those in 2017/18 (36.5; 95% CI = 33.5–39.0), representing a decline of 28.1%. When considering sex, boys were more active than girls in both surveys. While the prevalence of sufficient PA declined by 17.2% in boys in 2017/18 compared to 2007, in girls, this decline was significantly greater at 45.6% (OR = 0.59; 95% CI: 0.49–0.70). When analyzed by age, a declining trend over the age of 10 years was also observed, except for boys aged 15 and 18 years and girls aged 18 years. For these, the prevalence of sufficient physical activity remained stable.

In the present study, ORs for sufficient PA were estimated by comparing adolescents in 2017/18 to those in 2007 (Figure 1). In boys, there was no significant difference in sufficient PA in those aged 15 and 18 years. The opposite was observed in those aged 16 (OR = 0.57; 95% CI: 0.35–0.91), 17 (OR = 0.58; 95% CI: 0.36–0.94), and in total (OR = 0.68; 95% CI: 0.52–0.88), indicating that the chances of sufficient PA in adolescents in 2017/18 were significantly reduced. In girls, a trend similar to that of boys was also observed in practically all ages (except 18 years), showing that those in 2017/18 had reduced chances of being more active than those in 2007.

## 4. Discussion

The study revealed that the prevalence of sufficient PA declined over the 10-year interval in adolescents of both sexes, with a greater decline observed in girls. This trend was also observed in the analysis by age. The contribution of this study is proof that the secular decreases in PA as a result of sex and age meant that certain groups have been more exposed to the greatest variations in PA decreases over the last decade.

### 4.1. PA Declining Trend

Overall, the prevalence of sufficient PA declined considerably in this sample of Brazilian adolescents in the decade investigated, pointing to a challenging future. This scenario goes against the WHO Global Action Plan for Physical Activity, which set a target of a 15% increase in sufficient PA in adolescents worldwide by 2030 [1]. Despite this, the findings corroborate those of previous studies based on self-reported PA [7,12] and objectively measured PA [4,11], which found negative secular trends. Other studies showing positive trends [8,14,20] and stable trends [5,9] have also been found. These inconsistencies may be linked to the characteristics of the samples and the instruments used in the studies, as well as changes made to the protocols of the periods analyzed [12].

These divergences between studies may be linked to such factors as the presence or absence of public policies for the promotion of PA in the periods investigated and the comparison time used in each survey [15]. For example, in adolescents from China [21] and Brazil [9], stable trends in PA were found from data comparing surveys over six years. As for the studies that showed trends of increasing PA, a time equal to or greater than eight years of comparison was observed in adolescents from Europe and North America [4], as well as in those from Finland and Norway [8]. Among the studies showing declining trends, the comparison time used between surveys was nine years or more [4,7,10,12]. Therefore, it is assumed that there is a time necessary for changing intergenerational behavior. Thus, for those studies that did not find an upward or downward trend, it is likely that the sufficient time needed to achieve these changes had not been reached.

Regarding the declining trend observed in the present study, some hypotheses can be raised. In Brazil, the economic transition that occurred in the last decade marked by rapid urbanization and economic growth may, in part, explain this finding. These conditions favor fewer energy demands in the different domains of PA, whereas sedentary behaviors have been preferred during leisure time [22]. For example, a previous study conducted in the same region as the present study revealed changes in preferences for leisure activities in adolescents between 2001 and 2011 [23]. The authors showed that the preference for activities with higher energy expenditure decreased over time, while the preference for the use of screens (television, video games, and computers) increased. All of this seems to indicate that PA and sedentary behavior correlate differently with a country’s economic development [24].

### 4.2. Declining Trends in PA and Sex

Another particularly interesting finding was that the decline in sufficient PA was almost three times greater in girls than in boys. This result draws attention since, in the researched literature, it was possible to observe in adolescents from different regions of the world that PA did not decline in girls [3]. Despite this, declining trends, albeit to a lesser extent, have been observed in studies with Canadian [11], Portuguese [12], and Czech Republic adolescent girls [7]. This result may reflect the fact that adolescent boys and girls still reproduce patterns determined by a conservative society that strengthens sex inequalities [25]. From this perspective, PA can be seen as a behavior that would intensify the traits of masculinity in boys, while in girls, the constant engagement in PA would overshadow female traits.

Furthermore, assuming that girls have acquired more freedom of choice over time, they have traditionally been used to staying indoors, discrediting new opportunities for PA. Additionally, environmental factors, such as a lack of security and violence in public spaces for the practice of PA during leisure [26], may have reduced the interest of girls in exercising. In this case, they could spend more time at home with screen-based activities, such as using the computer [3]. A previous study based on data from Health Behavior in School-Aged Children (HBSC) revealed that, in the 30 countries investigated, the time spent using the computer over time increased for social and academic purposes only in girls [27]. These findings suggest that girls particularly need interventions and public policies to remain engaged in the PAs that they perform and mainly to adhere to new practices.

### 4.3. Declining Trends in PA and Age

The decline in sufficient PA appears to be age-dependent, especially in those aged 16 and 17 in 2017/17, whose chances for PA were reduced when compared to their 2007 peers. Most of the available evidence does not present results according to age or use different age amplitudes, making comparisons between studies difficult. For example, in Norway, secular trends in PA were based on information from 15-year-old adolescents [4], and in Finland, they were based on information from adolescents between 15 and 16 years old [28]. Another large study with information from 34 countries in Europe and North America examined the secular trends in adolescents aged 11 to 15 years [5], and in Portugal, the trend data was directed to adolescents aged 11 to 17 years [6]. However, our findings corroborate those of a previous study that showed a decline in preference for physical activities (−23.0%) and cultural activities (−12.6%) among adolescents 15 and 16 years old and an increased interest in electronic games (3.0%) and computer use (36.5%) between 2001 and 2011 [23]. It is believed that this trend of declining PA is linked to the new habits of adolescents who have easier access, in the vast majority of cases by parents, to screen devices (for example, smartphones) with multitasking. These devices have attracted the attention of adolescents, leading them to spend more time in sedentary behaviors, which tend to provide little or no time for PA [14].

### 4.4. Strengths and Limitations

The findings of this study need to be analyzed in light of some limitations. Initially, the use of self-report-based PA may have generated inaccurate results due to memory bias or the desire to provide socially desirable information [29]. Furthermore, the questionnaire used does not differ between the different domains (for example, leisure and active commuting), making it difficult to suggest further directions for decision making based on our findings. Economic level, which could add to the understanding of the results, was assessed in both surveys, but the instruments were changed. Thus, it was decided not to include such information due to the inherent bias in comparing information collected in different ways. Finally, the data presented does not represent adolescents across the country; those enrolled in private schools; and those who, for some reason, do not attend school. Among the strong points, the use of relatively large samples stands out. Comparable and repeated measures that followed the same protocol were followed at both times, which are essential prerequisites for studies of secular trends [7]. Finally, there was a comprehensive analysis of the outcome variable by sex and age subgroups.

## 5. Conclusions

The findings of the present study provide evidence that sufficient PA declined, rapidly, in this sample of Brazilian adolescents, and the lower prevalence among girls contributes, in large part, to this decline. Additionally, the results indicate that the declining trends were sex- and age-specific. Notably, girls aged 15 to 17 years in 2017/18 were the subgroups most exposed to declines in sufficient PA. Specific approaches for the promotion of PA in Brazilian adolescents must consider sex and age. Campaigns to promote PA in schools and in national media and the insertion of more forceful actions aimed at PA in the Brazilian National Health Promotion Policy can be interesting initiatives. For future studies on PA levels in Brazilian adolescents, we suggest a better methodology to minimize the risk of bias.

## Figures and Tables

**Figure 1 ijerph-19-16901-f001:**
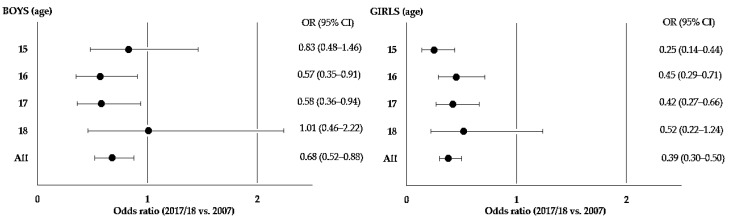
Age-specific odds ratios (ORs) with 95% confidence intervals (CIs) for sufficient PA in adolescents in 2017/18 compared with those in 2007. Florianopolis, SC, Brazil.

**Table 1 ijerph-19-16901-t001:** Descriptive characteristics of the samples, Florianopolis, Santa Catarina, Brazil.

Age (Years)	2007 n (%)	2017/18 n (%)
Boys		
15	119 (29.5)	85 (16.9)
16	138 (34.2)	143 (28.4)
17	112 (27.7)	188 (37.4)
18	35 (8.7)	87 (17.3)
All	404 (38.8)	503 (61.2)
Girls		
15	172 (27.0)	100 (21.1)
16	226 (35.5)	147 (30.9)
17	186 (29.2)	180 (37.9)
18	52 (8.2)	48 (10.1)
All	636 (61.2)	475 (48.6)
Socioeconomic status		
Low	385 (37.0)	352 (36.0)
Middle	351 (33.8)	308 (31.5)
High	304 (29.2)	318 (32.5)

**Table 2 ijerph-19-16901-t002:** Prevalence of sufficient physical activity in adolescents. Florianopolis, Santa Catarina, Brazil.

Age (Years)	2007 ^a^ Prevalence (95% CI)	2017/2018 ^b^ Prevalence (95% CI)	Variation (%) [(b − a)/a] × 100
Boys			
15	50.4 (40.2–57.0)	45.9 (36.3–53.1)	−8.9%
16	60.9 (53.9–67.4)	46.9 (40.5–50.3)	−23.0%
17	60.7 (53.8–65.9)	47.3 (42.4–51.7)	−22.1%
18	45.7 (29.6–59.0)	46.0 (36.6–53.0)	0.7%
All	56.4 (53.5–58.9)	46.7 (43.6–49.3)	−17.2%
Girls			
15	52.9 (45.4–59.3)	22.0 (14.0–28.0)	−58.4%
16	46.0 (40.4–50.3)	27.9 (22.0–32.0)	−39.3%
17	46.2 (40.3–50.6)	26.7 (21.6–31.2)	−42.2%
18	36.5 (24.1–47.1)	22.0 (13.5–30.3)	−39.7%
All	47.2 (44.2–49.7)	25.7 (23.1–28.0)	−45.6%

Note: 95% CI: 95% confidence interval. ^a^ Initial value; ^b^ final value.

## Data Availability

Data sharing not applicable.

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
