# Peer review of "Physical Activity Levels in Brazilian Adolescents: A Secular Trend Study (2007–2017/18)"

_ijerph, 2022, doi:10.3390/ijerph192416901_

Round 1

Reviewer 1 Report

Dear authors,

“Physical Activity Levels in Brazilian Adolescents: A Secular
Trend Study (2007-2017/18)” is well-written a manuscript about the change on physical activity levels and the importance of promotion of physical activity among adolescent Brazilian population. However, there are great limitations to not accept the article.

INTRODUCTION

-        Line 71: Objective. Could there be differences by region and economic level? This aspects are important because “Brazil is a country with continental dimensions, with different levels of economic equality, climate, and lifestyle. (Line 62-63)”. This subgroup analyses  are not included in results section.

METHOD

-        Line 90: “The school with the highest number of enrolments in each region was selected for data collection”

This choice directly affects the economic level, and this variable is not considered in the manuscript. In addition, this outcome was assessed with different instruments in different time points ( 2007 and 2017-18). It is essential to know the economic level for data consistency.

“There is an association between socioeconomic status (SES) and physical activity among adolescents, and that adolescents with higher SES are more physically active than those with lower SES” (https://doi.org/10.1111/j.1600-0838.2009.01047.x)

Future studies on physical activity levels in Brazilian Adolescent must propose better methodology and minimise risk of bias.

Author Response

Reviewer 1

Dear Reviewer,

We appreciate all the valuable suggestions previously made for the improvement of this manuscript. We adjusted the recommendations point by point as shown below. We also highlight the requested changes in the manuscript.

Best Regards,

--------------------------------------------------------------------------------------

INTRODUCTION

- Line 71: Objective. Could there be differences by region and economic level? This aspects are important because “Brazil is a country with continental dimensions, with different levels of economic equality, climate, and lifestyle. (Line 62-63)”. This subgroup analyses are not included in results section.

- We appreciate the comment. We reviewed the reference mentioned and included a justification for the present study based on the information presented.

METHOD

-Line 90: “The school with the highest number of enrolments in each region was selected for data collection”

This choice directly affects the economic level, and this variable is not considered in the manuscript. In addition, this outcome was assessed with different instruments in different time points ( 2007 and 2017-18). It is essential to know the economic level for data consistency.

-To meet the reviewer's request, we chose to include the economic status variable for the purposes of characterizing the adolescents, even though it was collected in a different way. However, we chose to treat it as a tertile since, in both instruments, the higher the total score, the higher the economic level. We still leave a note assuming the use of the variable as a limitation.

“There is an association between socioeconomic status (SES) and physical activity among adolescents, and that adolescents with higher SES are more physically active than those with lower SES” (https://doi.org/10.1111/j.1600-0838.2009.01047.x)

- See previous answer.

-Future studies on physical activity levels in Brazilian Adolescent must propose better methodology and minimize risk of bias.

-We appreciate the suggestion. We thought it appropriate to include the information in the study conclusions.

Reviewer 2 Report

Dear Author,

The manuscript titled as " Physical Activity Levels in Brazilian Adolescents: A Secular Trend Study (2007-2017/18) ". Comments and Suggestions for Authors

The paper is overall well structured despite a some comments. I have suggestion that could improve the paper:

-          Introduction

o    The introduction explains the most important information about the study.

-          Material and Methods

o    Please explain in statistical section: Prevalence of sufficient physical activity.

-          Results:

o    The table 2 is unclear. Please explain table 2 (add units).

-          Discussion

o    In my opinion the discussion is well written.

Kind regards

Author Response

Reviewer 2

Dear Reviewer,

We appreciate all the valuable suggestions previously made for the improvement of this manuscript. We adjusted the recommendations point by point as shown below. We also highlight the requested changes in the manuscript.

Best Regards,

--------------------------------------------------------------------------------------

INTRODUCTION

-The introduction explains the most important information about the study.

- We appreciate the comment.

- MATERIAL AND METHODS

-Please explain in statistical section: Prevalence of sufficient physical activity.

- Thanks for the suggestion. We added the requested information.

RESULTS:

-The table 2 is unclear. Please explain table 2 (add units).

- Thanks for the suggestion. We added the requested information.

DISCUSSION

In my opinion the discussion is well written.

- We appreciate the comment.

Reviewer 3 Report

Thank you for your submission. The manuscript reports adolescent PA levels in 2007 and again in 2018/18 (examining the trends over 10 years). The results of the study showed that both boys and girls were more active in 2007 than their peers in 2017/18. This manuscript is reasonably clear and concise but needs some revisions to grammar and English sentence structure. The authors have done a good job of providing an informative and meaningful addition to the current study field.

However, there are several changes that the authors are encouraged to revise to elevate the overall contribution of the paper to this research field.

Abstract 

- "How-ever"? - consider restructuring this sentence

- should you mention IPAQ in the abstract?

Introduction

- good opening paragraph - presents a problem

- lines 45-47 - difficult to understand

- I think reference 8 is a tough comparison given it begins in 1985

- what was your hypothesis?

- why examine 2007-2017/18? Or is it because of the survey mentioned at the beginning of the methods section 2.1?

Methods

- why do you report it as 2007-2017 or 2018 if the time frame is 10 years?

- what if a participant completed 45 min of PA on 6 occasions throughout the week? would it count at all? (Trying to figure out the 60 min of MVPA info)

- p in italics - p

Discussion

- first sentence is missing something.

- very nice breakdown and thoroughness 

Conclusions 

- I think it would be good to include a more practical application suggestion (e.g., how might Brazilians health authorities do something about these findings?) 

Author Response

Reviewer 3

Dear Reviewer,

We appreciate all the valuable suggestions previously made for the improvement of this manuscript. We adjusted the recommendations point by point as shown below. We also highlight the requested changes in the manuscript.

Best Regards,

--------------------------------------------------------------------------------------

Abstract

- "How-ever"? - consider restructuring this sentence

-Thanks for the observation. Adjustment performed.

- should you mention IPAQ in the abstract?

-Thanks for the observation. Adjustment performed.

Introduction

- good opening paragraph - presents a problem

-We appreciate the comment.

- lines 45-47 - difficult to understand

-We appreciate your observation. We rewrote the sentence.

- I think reference 8 is a tough comparison given it begins in 1985

- We appreciate the comment. We decided to keep the citation due to the difficulty in finding studies that performed analyzes by age.

- what was your hypothesis?

- We added our hypothesis in the last paragraph of the introduction.

- why examine 2007-2017/18? Or is it because of the survey mentioned at the beginning of the methods section 2.1?

-Unfortunately, we do not have data from other periods, and as we had a baseline with data from 2007, we decided to replicate the survey 10 years later (2017).

Methods

- why do you report it as 2007-2017 or 2018 if the time frame is 10 years?

-Thanks for your comment. We forgot to inform in the methods that it was necessary to extend the collection to 2018 for logistical reasons. Information has been added.

- what if a participant completed 45 min of PA on 6 occasions throughout the week? would it count at all? (Trying to figure out the 60 min of MVPA info)

-We appreciate your comment. We defend the idea that the amount of AF, in any amount, is important at all stages of life. We adjusted the writing to make the study's interest in the World Health Organization's recommendation clear.

- p in italics - p

-Adjustment performed.

Discussion

- first sentence is missing something.

-Adjustment performed.

- very nice breakdown and thoroughness

-We appreciate the comment.

Conclusions

- I think it would be good to include a more practical application suggestion (e.g., how might Brazilians health authorities do something about these findings?)

- We appreciate the suggestion. A statement has been included in the conclusion.

Round 2

Reviewer 1 Report

Dear Authors,

I considered the paper is prepared for publication. I like the comment on economic transition that occurred in the last decade marked by rapid urbanization and economic growth. Congrats.